# Amplectobeluid Radiodont *Guanshancaris* gen. nov. from the Lower Cambrian (Stage 4) Guanshan Lagerstätte of South China: Biostratigraphic and Paleobiogeographic Implications

**DOI:** 10.3390/biology12040583

**Published:** 2023-04-11

**Authors:** Mingjing Zhang, Yu Wu, Weiliang Lin, Jiaxin Ma, Yuheng Wu, Dongjing Fu

**Affiliations:** 1State Key Laboratory of Continental Dynamics, Department of Geology, Northwest University, Xi’an 710069, China; 2Shaanxi Key Laboratory of Early Life and Environment, Department of Geology, Northwest University, Xi’an 710069, China

**Keywords:** Panarthropoda, *Guanshancaris* gen. nov., frontal appendage, durophagous predation, biogeographic patterns, habitat preferences

## Abstract

**Simple Summary:**

“*Anomalocaris*” *kunmingensis* is the most common radiodont in the Guanshan biota, which has been recently reassigned to the family Amplectobeluidae. However, its generic assignment is still uncertain. Our new specimens reveal more characteristic of “*Anomalocaris*” *kunmingensis*, warranting the erection of a new genus, *Guanshancaris* gen. nov. Brachiopod shells bearing embayment injury, and broken trilobites closely associated with the frontal appendages, may indicate durophagous habits of *Guanshancaris*. The spatial and temporal distribution of Amplectobeluidae, documented in various soft-bodied biota, indicates that this group was restricted to low latitude regions from Cambrian Stage 3 to Drumian, and implies an ecological preference for a shallow-water environment.

**Abstract:**

Radiodonta, an extinct stem-euarthropod group, has been considered as the largest predator of Cambrian marine ecosystems. As one of the radiodont-bearing Konservat-Lagerstätten, the Guanshan biota (South China, Cambrian Stage 4) has yielded a diverse assemblage of soft-bodied and biomineralized taxa that are exclusive to this exceptional deposit. “*Anomalocaris*” *kunmingensis*, the most abundant radiodont in the Guanshan biota, was originally assigned to *Anomalocaris* within the Anomalocarididae. Despite this taxon being formally assigned to the family Amplectobeluidae more recently, its generic assignment remains uncertain. Here, we present new materials of “*Anomalocaris*” *kunmingensis* from the Guanshan biota, and reveal that the frontal appendages possess two enlarged endites; all endites bear one posterior auxiliary spine and up to four anterior auxiliary spines; three robust dorsal spines and one terminal spine protrude from the distal part. These new observations, allied with anatomical features illustrated by previous studies, allow us to assign this taxon to a new genus, *Guanshancaris* gen. nov. Brachiopod shell bearing embayed injury and incomplete trilobites, associated with frontal appendages in our specimens, to some extent confirm *Guanshancaris* as a possible durophagous predator. The distribution of amplectobeluids demonstrates that this group is restricted to Cambrian Stage 3 to Drumian, and occurs across South China and Laurentia within the tropics/subtropics belt. Moreover, the amount and abundance of amplectobeluids evidently decreases after the Early–Middle Cambrian boundary, which indicates its possible preference for shallow water, referring to its paleoenvironmental distribution and may be influenced by geochemical, tectonic, and climatic variation.

## 1. Introduction

Radiodonts, a large soft-bodied stem group of euarthropods [1,2] with a cosmopolitan distribution from Cambrian to Ordovician, have been viewed as giant apex predators [3,4] in early Paleozoic marine ecosystems. Radiodonts are often preserved as disarticulated elements in radiodont-bearing Konservat-Lagerstätten, such as Emu Bay Shale (Australia), Kinzers Formation (USA), Valdemiedes Formation (Spain), Balang Formation (China), and Guanshan biota (China), of which the frontal appendage is the most commonly preserved part [5,6,7,8,9,10]. With considerable morphological disparity, the frontal appendages of radiodonts are suggestive of a range of feeding modes to exploit diverse prey resources, roughly including apex predation, generalized predation, and filter feeding (e.g., [3,11,12,13,14,15,16,17,18,19,20]). Hence, morphology of frontal appendages is an essential trait to classify radiodonts into four families, including Anomalocarididae [21], Amplectobeluidae [22], Tamisiocarididae [8], and Hurdiidae [23]. Amplectobeluidae is characterized by proximal endite with slightly large to hypertrophied size, and gnathobase-like structures [22,24,25]. Based on the many-segmented appendages with a wide range of mobility, and the elongated endite supporting a scissor-like gripping motion [22,25,26,27], amplectobeluids are able to grasp the prey and manipulate them towards the mouth with the help of frontal appendages.

The Guanshan biota, a classic Burgess Shale-type Konservat-Lagerstätten occurring in the Wulongqing Formation in eastern Yunnan of South China, preserves a diverse assemblage of exceptionally preserved body fossils [28]. This exceptionally preserved biota is confined to Stage 4 in Cambrian, coeval to the Emu Bay Shale biota of South Australia [29], the Kinzers biotas of Laurentia [30], and the Sinsk biota of Siberia [31], which links the older Chengjiang biota (Cambrian Series 2, Stage 3) and the younger Kaili biota (Cambrian Miaolingian Series, Wuliuan Stage) in South China [27,32]. Therefore, the Guanshan biota has an indispensable role to illuminate the history of the early evolution of marine animals during the ‘Cambrian Explosion’. To date, approximately 60 taxa belonging to more than 10 metazoan groups and green algae have been described in Guanshan [28,33], such as chancelloriids, cnidarians, brachiopods [34], hyoliths, eldonioids, priapulids [35], lobopodians, eocrinoid echinoderms [36], vetulicoliids [37], as well as arthropods [38,39,40,41,42].

Since the original description of radiodont material from the Guanshan biota, radiodonts are represented by at least four taxa, containing “*Anomalocaris*” *kunmingensis*, *Paranomalocaris multisegmentalis*, *P. simplex*, and an unnamed tamisiocaridid species [7,10]. “*Anomalocaris*” *kunmingensis* is of the most abundance radiodonts in the Guanshan biota, which was originally assigned to *Anomalocaris* [7]. A recently reported well-preserved radiodont oral cone was tentatively considered to be “*Anomalocaris*” *kunmingensis*, owing to its great abundance in Guanshan [43]. Although “*Anomalocaris*” *kunmingensis* has always been established as a member of Amplectobeluidae in numerous phylogenetic analyses over the past decade (e.g., [16,17,23,44]), until most recently, this taxon was formally assigned to the family Amplectobeluidae [10]. By far, however, this taxon still belongs to an uncertain genus. In this contribution, the discovery of new morphological features based on additional well-preserved specimens lead us to redescribed “*Anomalocaris*” *kunmingensis* in the context of the new generic assignment, and confirm its amplectobeluid affinity. Additionally, our specimens may also display the durophagous eating habits of amplectobeluids and a predator–prey relationship of *Guanshancaris* and other species, such as brachiopods and trilobites. Our work herein also adds to spatio-temporal distributional data on amplectobeluid radiodonts, and comments on paleogeographic distribution patterns and habitat preferences of this group.

## 2. Materials and Methods

The fossil materials described in this contribution were excavated from the Gaoloufang (prefix GLF) section (24°57′36″ N, 102°48′15″ E) at Guangwei in Kunming, Yunnan Province, China, one of the main localities of the Guanshan Lagerstätte (Cambrain Series 2, Stage 4) (Figure 1; [28,33]). For detailed comparison, we also examine two specimens of *Laminacaris* (SJZ B02 454B and SK2211B) from two localities of the Cambrian (Series 2, Stage 3) Yu’anshan Member of the Chiungchussu Formation (the Chengjiang Lagerstätte) in eastern Yunnan Province, China; namely, Sanjiezi (prefix SJZ) section (25°06′34″ N, 102°84′73″ E) in Erjie town at Jinning county, and Shankoucun (prefix SK) section (24°49′49″ N, 102°24′55″ E) at Anning county. All specimens are housed at the Shaanxi Key Laboratory of Early Life and Environments (LELE) and Department of Geology, Northwest University (NWU), Xi’an, China.

All specimens were gathered from yellowish-to-greenish mudstones interbedded by thin-bedded siltstone and sandstone [33,43,46]. Some were further prepared with fine needles under high magnification using stereomicroscopes. Fossils were photographed with a Canon EOS 5D Mark II digital camera with a Canon EF–S 60 mm macro lens (general view) and an EF–S 80 mm micro lens (detailed view), and was controlled using the EOS Utility 3.2 program for remote shooting. Images were processed in Digital Photo Professional 4. Camera lucida drawing were made using a Zeiss Discovery V12 microscope and prepared with CorelDRAW Graphics Suite 2022.

The descriptive terminology of the frontal appendage mainly follows that of Lerosey-Aubril and Pates [23] and Guo et al. [47]. Radiodont frontal appendages can be separated into two major regions for amplectobeluids, anomalocaridids and tamisiocaridids, namely, peduncle and distal articulated region. The term ‘peduncle’ [48] (also called a ‘base’ sensu [49]) is used to describe proximal podomeres of frontal appendages, which have more weakly defined articulations and bear no endite or reduced endite; the ‘distal articulated region’ (defined by Cong et al. [25]; equivalent to ‘claw’ sensu [49]) refers to the distal part of appendages, where endites project from the ventral side of podomeres and may bear sophisticated auxiliary spines. An angle on dorsal margin can normally be identified as the boundary between the peduncle and the distal articulated region. The term ‘endite’ is employed to describe the ventral spinose or setose projection on podomeres, and the term ‘auxiliary spines’ refers to spines projecting from the endite.

## 3. Results

### Systematic Paleontology

Superphylum PANARTHROPODA Nielsen, 1995 [50].(Stem–group) ARTHROPODA Von Siebold, 1848 [51].Order RADIODONTA Collins, 1996 [26].Family AMPLECTOBELUIDAE Pates, Daley, Edgecombe, Cong, Lieberman, Zhang, 2019 [22].Genus *Guanshancaris* gen. nov.

Etymology. From the name of the Guanshan biota containing the new genus.

Type species. *Anomalocaris kunmingensis* Wang et al., 2013 [7]; Gaoloufang village, Kunming, Yunnan, China, Wulongqing Formation (*Palaeolenus* and *Megapalaeolenus* zones), Cambrian (Stage 4).

Diagnosis. Amplectobeluid with frontal appendages consisting of 15 podomeres, including 2 peduncular podomeres and 13 distal articulated podomeres; podomeres tall rectangle in shape; endites on the proximal-most podomere in the distal articulated region and distal-most peduncular podomere enlarged, with the latter being slightly smaller than the former; proximal-most endite in the distal articulated region possesses four anterior auxiliary spines, increasing in length towards the tip of the endite; proximal endites distal to the proximal-most one bear up to three anterior auxiliary spines; distal endites bear paired auxiliary spines; all endites bear one posterior auxiliary spine projecting from the middle position; three thickened dorsal spines protrude from the distal-most three podomeres and one terminal spine on the distal-most podomere.

Stratigraphic and geographic range. Gaoloufang village and Lihuazhuang section, Yunnan, China, Wulongqing Formation (*Palaeolenus* and *Megapalaeolenus* zones), Cambrian (Stage 4; c. 5.10~5.15 Ma).

Remarks. *Guanshancaris kunmingensis* was originally assigned to *Anomalocaris* based on the number of podomeres and alternation in the length of the endites on odd/even numbered podomeres (Figure 1 in [7]), which are shared with typical *Anomalocaris* species (Figures 1–7 in [52]). This taxon was later generally retrieved as a basal member of Amplectobeluidae in several phylogenetic studies (e.g., [16,17,23,44,53]), and thus was most recently formally assigned to the Amplectobeluidae [10]. However, this Guanshan taxon was still simply treated as either *Anomalocaris kunmingensis* or “*Anomalocaris*” *kunmingensis*, lacking a formal generic name [10]. In this contribution, we established a new genus, *Guanshancaris* gen. nov., for this taxon within the family Amplectobeluidae.

*Guanshancaris* can be confidently identified as Amplectobeluidae, as presently defined, by the diagnostic features of the frontal appendage, including an elongated rectangle shape of the podomeres, the largest enlarged proximal-most endite in the distal articulated region, the endite on podomere five is larger than podomere three (counting from posterior to anterior end) in the distal articulated region, as well as the thickened dorsal spines and terminal spine in the distal region. A recently reported amplectobeluid frontal appendage from the Parker Formation resembles *Guanshancaris* in many regards, such as 13 podomeres in the distal articulated region; two enlarged endites, respectively, on the distal-most peduncular podomere and proximal-most podomere in the distal articulated region; and three anterior auxiliary spines protruding from the proximal-most podomere at an oblique angle in the distal articulated region [54] (p. 778). Hence, here we consider that the appendage from the Parker Formation may represent the species of *Guanshancairs* outside the Guanshan biota.

In the context of Amplectobeluidae, *Guanshancaris* uniquely possesses relatively large peduncular endites, as well as larger and stouter endites distal to the enlarged proximal-most one in the distal articulated region (Figure 2; Table 1). The condition that the endites of *Guanshancaris* bears auxiliary spines differentiates it from *Amplectobelua*, in which the spiniform endites are generally devoid of auxiliary spines (Figure 15 in [55] and Figure 4 in [24]). *Guanshancaris* also differs from *Amplectobelua* by the length of the proximal-most enlarged endite in the distal articulated region less than one-third of the length of the appendage, whereas the exceptionally long endite is one-third to nearly half as long as the length of the appendage in *Amplectobelua* [24,55]. The enlarged endite in *Guanshancaris* is not inclined to the distal end as much as *Amplectobelua symbrachiata* from the Chengjiang biota (Figure 4 in [24]) and *Amplectobelua stephenensis* from the Burgess Shale (Figure 2A,C,D; text—Figure 4 in [12]). *Guanshancaris* differs from *Lyrarapax* and *Ramskoeldia* in the number of podomeres. The appendage of *Guanshancaris* possesses 15 podomeres, whereas *L. trilobus* and *Ramskoeldia* bear 11 podomeres (Figure 3.1 in [56]) and 16 podomeres, respectively (Figures 2 and 4 in [25]). Unlike *Lyrarapax* and *Ramskoeldia*, the podomeres of *Guanshancaris* have a higher height/length ratio (approximatively ~2.0).

*Guanshancaris kunmingensis* (Wang et al., 2013) comb. nov.2013 *Anomalocaris kunmingensis* sp. nov.; Wang et al., 2013, pp. 3938–3939, Figure 1 [7].2013 *Anomalocaris kunmingensis*; Hu et al., 2013, Figures 178–180 [28].2014 *Anomalocaris kunmingensis*; Cong et al., 2014, ext. data Figure 4 [53].2014 *Amplectobelua kunmingensis*; Vinther et al., 2014, Figure 3 [16].2015 *Anomalocaris kunmingensis*; Van Roy et al., 2015, ext. data Figure 10 [17].2018 *Anomalocaris kunmingensis*; Lerosey-Aubril and Pates 2018, Figure 2, sup. Figures 4 and 5 [23].2018 *“Anomalocaris” kunmingensis*; Liu et al., 2018, sup. Figures 2 and 3C [44].2017 Radiodontan gen. indet. sp. indet.; Zeng et al., 2017, Figure 2 [43].2021 *“Anomalocaris” kunmingensis*; Jiao et al., 2021, pp. 260–262, Figures 2–4 [10].

Etymology. From the combination of the new generic name *Guanshacaris* and the species name of “*Anomalocaris*” *kunmingensis*.

Type material. Holotype: GLF WLQ 02, part and counterpart, preserve a pair of nearly complete frontal appendage; paratypes: GLF WLQ 05; GLF WLQ 10, part of isolated frontal appendage.

Type locality. Gaoloufang village, Kunming, East Yunnan, China [7].

Type horizon. Wulongqing Formation (*Palaeolenus* and *Megapalaeolenus* Zone), Cambrian (Series 2, Stage 4) [7].

Materials. Five isolated appendage specimens, GLF WLQ 03, GLF WLQ 04, GLF WLQ 07, GLF WLQ 08, GLF WLQ 09.

Diagnosis. As for genus.

Occurrence. Gaoloufang section, Wulongqing Formation (*Palaeolenus* and *Megapalaeolenus* Zone), Cambrian (Stage 4, Series 2), Gaoloufang village, Kunming, eastern Yunnan, China.

Description. The frontal appendage of *Guanshancaris kunmingensis* has an elongated shape tapering towards the distal end. Lengths of the complete appendages range from 25.1 mm to 81.9 mm (measuring along dorsal margin). The most complete appendage specimen consists of 15 podomeres, containing 2 podomeres in the peduncle (p), which attach to 13 podomeres in the distal articulated region (po) at an angle of approximately 160° along the dorsal margin (Figure 3B,D and Figure 4H,J). The configuration of the proximal podomeres is elongated rectangular, and has a gradual transition to trapezium towards the distal end, with some tapering in width ventrally (Figure 3A–D and Figure 4A,B). The height/length ratio of the podomeres is approximately ∼2.0. Endites join to the ventral surface at an oblique angle of equal or less than 90°, of which the length alternates on the distal-most peduncular podomere (p1) and podomeres in the distal articulated region (po1–12) (longer ones on odd podomeres). Although most of the endites are not completely preserved, the length of the endites is approximately equal or exceeds the height of the attached podomeres in light of the width of the preserved base of the endites. The length of the endites on p1 and po1–12 diminishes towards the distal end, with the exception of the endite on po5, which is longer than the endite on po3 (Figure 3A–D and Figure 5A,B). Two specimens of our material are smaller than other specimens, and the articulations between podomeres are not observable, which represent possible juvenile individuals (Figure 5).

The proximal-most peduncular podomere is weakly sclerotized (p2 in Figure 3A–D and Figure 4A,B), which may bear an endite inclining towards the distal end (pe2 in Figure 4H,J,K). The endites of p1 and po1 are larger and stouter than other endites, and the latter is thicker (Figure 3B,D and Figure 4A,B,H,J,K). The endite on p1 is not preserved completely in our specimens, which can be seen in one posterior auxiliary spine in WLQ GLF 07 (Figure 4K) and one possible anterior auxiliary spine in the holotype (Figure 3B,D). The endite on po1 is flanked by one large posterior auxiliary spine from nearly midpoint of the endite, and up to four anterior auxiliary spines, which increase in size towards the end of the endite (Figure 3G, Figure 4K and Figure 5A,B). Endites on proximal podomeres distal to po1 bear one posterior, and up to three anterior, auxiliary spines (Figure 3A–D and Figure 5A–D). One pair of auxiliary spines projects from the endites on distal podomeres (Figure 5A–D). All the auxiliary spines form an acute angle with the endites (Figure 3 and Figure 4). Three arched robust dorsal spines project from the distally dorsal margin of po11–13 (Figure 3H,I). Some isolated fragments by the distal side of the dorsal margin are possibly dorsal spines (Figure 3H,I and Figure 6A,B). A terminal spine curved towards the ventral surface of the appendage is present on po13 (Figure 3A–D,H,I). Distinct triangular flexible membranous areas between podomeres can be seen in GLF WLQ 10 (Figure 3H,I).

An oval and an elongated carapace are associated with the frontal appendage in GLF WLQ 04 (Figure 5A,B), which may represent the central head shield and lateral P-element of *Guanshancaris kunmingensis*; the oval head shield measures 12.3 mm from the overlapping edge to the incomplete edge at its furthest point, and 9.6 mm at its widest point; the lateral P-element is 13.4 mm in length and 4 mm in width separately. The P-element is composed of two parts, which overlap each other. The oval head shield is not entire, and is partly overlapped by the P-element on one side. Both of these carapaces do not show a reticulate pattern. Noticeably, the shells of brachiopods, the cephalic and thoracic regions of trilobites, as well as unidentified fragments, are located immediately adjacent to the frontal appendages in most of our specimens (Figure 3A–D, Figure 4A,B,G–J and Figure 6A,B). In GLF WLQ 09, one brachiopod shell has a concave embayed fracture in its top-left section, which clamps between the two dorsal spines of a frontal appendage (Figure 6A–C). The estimated maximum shell length, as well as width, is 6.0 mm and 4.2 mm, respectively; the fracture embayment in this shell is 0.96 mm in width (Figure 6D,E). Two sets of adjoining drape-like arcs can be seen in the embayment, which is wrapped around by a U-shaped relief (white arrows in Figure 6D,E). Although the co-occurrence of frontal appendages with trilobites and brachiopod shells does not indicate their predator–prey relationship directly, the existence of the brachiopod with the embayment injury, associated with the frontal appendage of *Guanshancaris*, increases the possibility of their predatory relationship.

Stratigraphic and geographic range. Gaoloufang village and Lihuazhuang section, Yunnan, China, Wulongqing Formation (*Palaeolenus* and *Megapalaeolenus* zones), Cambrian (Stage 4).

Remarks. *Guanshancaris kunmingensis*, which was originally reported by Wang et al. [7], is the most common radiodont taxa in the Guanshan biota. In their study, p1 has a tiny endite and one or two pairs of auxiliary spines present on the endites of po4–14 in the frontal appendage of *G. kunmingensis* [7]. In our specimens, p1 has an enlarged endite, and the endites on podomeres in the distal articulated region are flanked by one posterior auxiliary spine, and up to four anterior auxiliary spines (Figure 3A–D and Figure 5A–D). Furthermore, the enlarged endite on the distal-most peduncular podomere can also be seen in Wang et al. [7], which does not receive their attention, but has been mentioned in Jiao et al. [10] (p. 259). An oral cone with tetraradial configuration in Guanshan is described by Zeng et al. [42] from the Gaoloufang section. Due to the great abundance of frontal appendages of *G. kunmingensis* in this section, they deduce that this tetraradial oral cone may be *G. kunmingensis*. Recently, a pair of frontal appendages of *G. kunmingensis,* associated with a tetraradial oral cone bearing small and large plates, was reported from the Lihuazhuang section of the Guanshan biota (Figures 2–4 in [10]), which confirms the presence of a tetraradial oral cone in *G. kunmingensis*.

*Laminacaris chimera*, a species with combination of characters shared by hurdiids and other radiodont families, was reported in the Chengjiang biota by Guo et al. [47]. Frontal appendages of *L. chimera* have 15 podomeres and 2 large endites on p1 and po1, which show similarity with *G. kunmingensis* (Figure 2A,B and Figure 4C–F; Table 1). However, the podomeres have a higher height/length ratio (approximatively ~2.0) in *G. kunmingensis* than *L. chimera*. In *L. chimera*, each podomere bears a single endite [47], whereas the endites are paired on the podomeres of *G. kunmingensis*. The length of the endites on podomeres 4–14 is shorter than the height of the attached podomeres (Figures 1 and 2A–D in [47]) in *L. chimera*, whereas in *G. kunmingensis*, the endites are approximately equal or longer than the podomeres to which they attach (Figure 3A–D and Figure 5A–D). The shape of the endites on p1 and po2 in *L. chimera* is straight blade-like, and resembles the morphology of the endites of hurdiids (Figure 4F), which is not observable in *G. kunmingensis*. The endite on po2 bears small auxiliary spines between the larger auxiliary spines, and all the auxiliary spines arranged perpendicular to the endites in *L. chimera* (Figures 1H and 2E in [47]). In *G. kunmingensis*, the morphology of the endites on po2 distinctively differs from *L. chimera* by one large posterior auxiliary spine, and up to four anterior auxiliary spines protruding from the endite with a certain obliquity, which gradually become larger towards the end of the endite (Figure 3G, Figure 4K and Figure 5A,B).

## 4. Discussion

### 4.1. Guanshancaris as a Possible Durophagous Predator

As a member of Amplectobeluidae, *Guanshancaris* tends to use a variety of feeding structures, such as frontal appendages and oral cone, to consume biomineralised prey, which is represented by trilobites and brachiopods [57,58]. That is, morphofunctional information in *Guanshancaris* may identify its durophagous eating strategy. The endites apparatus of the frontal appendages in *Guanshancaris* is distinct from other amplectobeluids: two large proximal endites of *Guanshancaris* are not as stout and large as *Amplectobelua* and *Lyrarapax*, however, they may make it more flexible for *Guanshancaris* to grasp prey; the endites in the distal articulated region are longer and stouter than other amplectobeluids, which is beneficial for smashing shells powerfully. These functional morphologies may complement the inability of the oral cone to produce certain injuries to trilobite exoskeletons or brachiopods shell [1,3,13]. It is noteworthy that similar frontal appendage structure in mature large individuals with complete auxiliary spines also exists in some small ones (Figure 5A–D), possibly reflecting raptorial feeding habits in juveniles. Specimens of the Burgess Shale arthropod *Sidneyia inexpectans,* with fragmented juvenile trilobites within the gut tract, show that some gnathobase-bearing arthropod taxa may consume exoskeletons [59]. Although it is not discovered in Guanshan yet, in Amplectobeluidae, the gnathobase-like structure used to manipulate and masticate food items is a diagnostic trait [22], so it may reveal the generality of durophagous feeding in this group. Moreover, the oral cone that potentially belongs to *Guanshancaris* possesses many scale-like nodes on the surface (Figure 2 in [20]), which implies that this feeding structure is likely to be more sclerotized than previously thought. Therefore, *Guanshancaris*, the only representative of amplectobeluids in the Guanshan biota, is most likely to be a durophagous animal.

The presumed durophagy of *Guanshancaris* is also, to some extent, supported by the evidence of the durophagous shell-breaking on a brachiopod shell (Figure 6). Embayment is a kind of scar shape produced by Cambrian durophagous predators on brachiopods (Figure 6C–E; [60,61]). Adjoining drape-like arcs (white arrows in Figure 6D,E) may show self-repairability of the brachiopod after the sublethal predation, which was previously reported in the Guanshan biota (Figure 2 in [60]). In our specimen, the brachiopod shell bearing repaired damages, stuck in the middle of two dorsal spines of *Guanshancairs,* may imply a predator–prey relationship between them (Figure 6). Among all known animals in the Guanshan biota, such injury on brachiopod shells is most likely caused by radiodonts in the act of predation [57,58,60,61,62]. Compared to other Guanshan radiodonts, the morphology of *Guanshancaris* appendages, such as the enlarged proximal endites and relatively stout dorsal spines, is best suited to grasping and attacking hard prey. Therefore, although this co-occurrence of frontal appendage and injured brachiopod shell reflects the predator–prey relationship with little possibility between the two individuals, it is likely that this relationship does exist between the two species. With a great abundance in Guanshan, trilobites may serve as another food source for *Guanshancaris*. A number of trilobite-dominated aggregates in Emu Bay Shale [14], and elliptical aggregates containing exoskeletal remains of bivalved arthropods, hyoliths, and trilobites in Maotianshan Shale [58], are interpreted as coprolites, which may be excreted by radiodonts based on the size of the coprolites. Nevertheless, more direct evidence of the durophagy of *Guanshancaris* is awaiting the further discovery of gut content and coprolites of this animal.

In summary, the durophagy of specific radiodont taxa has been discussed for a long time [14,57,58,61,62]. Here, we add new evidence of *Guanshancaris* on the basis of two aspects: the functional morphology of the frontal appendages and a brachiopod shell with preingestive breakage associated with frontal appendage. The weight of the evidence reviewed above leads us to assume that *Guanshancaris*, the most common radiodont in Guanshan, may be a durophagous predator that is likely able to attack brachiopods and trilobites using its frontal appendages.

### 4.2. Paleoenvironmental Distribution

Amplectobeluids documented in various Cambrian Konservat-Lagerstätten provide further evidence to suggest that this group has an ecological preference for the shallow-water environment (Table 2). *Ramskoeldia*, *Lyrarapax*, and *Amplectobelua symbrachiata* are nearly restricted to the Chengjiang biota, which deposits a shallow-water setting of the epeiric platform, with fossils either autochthonous or parautochthonous [63,64]. *Amplectobelua symbrachiata* also occurred in the Niutitang Formation of Guizhou, South China, which deposited in offshore shelf-basin facies of the Yangtze Platform [65]. A possible *Ramskoeldia consimilis* found outside of South China occurred in the Latham Shale, which is generally considered as having been deposited in a proximal shelf above the fair-weather wave [22]. A recently reported amplectobeluid of indeterminable species from the Fandian biota in South China was deposited in a similar setting with the Chengjiang biota [66]. *Guanshancaris* is preserved in the Guanshan biota, deposited in an offshore transition between the fair-weather wave base and the storm wave base [33,46]. *Amplectobelua* is a widely distributed species of amplectobeluids, which occurs from the outer shelf adjacent to a carbonate ramp (Wheeler Formation, UT, USA, [67]; Burgess Shale, British Columbia, Canada, [3]; Kinzer Formation, PA, USA, [8]) to a lower shoreface setting (Chengjiang biota, Yunnan, South China, [64]; Niutitang Formation, Guizhou, South China, [65,68]). Although the Kinzer Formation, where an affinis species of *Amplectobelua symbrachiata* occurred [8], was deposited in a low-energy deep environment with intermittent, pulsed sedimentation, the sediment of this deposit may be transformed from further shallow inboard or elsewhere [30]. A few amplectobeluid taxa appear to be restricted to deep-water environments, as exemplified by *Amplectobelua stephenensis* from the Miaolingian Burgess Shale (Canada) and Wheeler Formation (USA) of Laurentia [3,69], and putative *Guanshancaris* from the Parker Quarry Lagerstätte (Franklin Basin succession of Parker Formation, northwest Vermont, USA, [56,70]). Considering both the diversity and abundance of amplectobeluids, however, it appears to be extremely rare in these three deposits (Table 2). The relationship between the changes of amplectobeluid species abundance and the depositional environment of assorted deposits may imply that this group is more suitable for shallow-water environments. With adequate sunlight and oxygen, the shallow-water setting was a preferable place for prey such as trilobites to live, which may provide a sufficient food for amplectobeluids.

### 4.3. Spatio–Temporal Distribution of Amplectobeluidae

During the Early and the Middle Cambrian (Series 2 and Miaolingian), four genera of the family Amplectobeluidae have been reported from distinct paleocontinents. The biogeographical and temporal distribution of Amplectobeluidae is summarized in Figure 7.

In the Cambrian Stage 3, records of amplectobeluids are relatively abundant, but the present state of knowledge is limited to the South China paleocontinent (eastern Yunnan), including *Amplectobelua symbrachiata* from the Zunyi biota (Niutitang Formation, *Tsunyidiscus niutitangensis* Zone) of Guizhou [65], *Amplectobelua symbrachiata*, *Ramskoeldia platyacantha*, *R. consimilis*, *Lyrarapax unguispinus,* and *L. trilobus* from the Chengjiang biota of Yunnan [24,25,44,53,55,56], as well as an undetermined amplectobeluid taxon from the Fandian biota of Sichuan (Yuxiansi Formation, *Yilangella–Zhangshania* Zone) [66]. During this period, amplectobeluids are not known from other main paleocontinents, such as Laurentia, North China, Avalonia, and Gondwana (Figure 7B).

Amplectobeluids appear to decline in both diversity and abundance in Stage 4, but have a wider distribution longitudinally, occurring across Laurentia and South China (Figure 7). *Guanshancaris kunmingensis* from the Guanshan biota (*Palaeolenus* and *Megapalaeolen* Zone; Figure 2, Figure 3, Figure 4 and Figure 5; [7,10]) constitutes the only representative of the family Amplectobeluidae from South China during this period. The amplectobeluids in Laurentia are represented by *Amplectobelua* aff. *symbrachiata* from the Kinzers Formation (*Bonnia*-*Olenellus* Zone) of Pennsylvania [8], a potential *Guanshancaris* species from the Parker Quarry Lagerstätte (*Bolbolenellus euryparia* Zone or overlying *Nephrolenellus multinodus* Zone) of Vermont [54,69], and *Ramskoeldia consimilis*? from the Latham Shale (*Olenellus* Zone, *Bristolia* Subzone) of California, USA [22,72].

During the middle Cambrian (Miaolingian Series), records of amplectobeluid continued to be sparse and apparently limited to Laurentia, including *Amplectobelua stephenensis* from the Burgess Shale of Canada (Latest *Glossopleura* Zone to early–middle *Bathyuriscus*-*Elrathina* Zone; [12]), and *Amplectobelua* cf. *A. stephenensis* from the Wheeler Formation (Drumian Stage) of Utah, USA (*Bolaspidella* trilobite Zone, *Ptychagnostus atavus agnostoid* Zone; [69]), which represents the youngest fossil record of amplectobeluids. Ultimately, this group virtually disappeared from the upper Miaolingian rock record (Figure 7A).

The apparent decline of amplectobeluids after Cambrian Stage 4 may be related to the tectonic, climate, and geochemical changes at the Early–Middle Cambrian boundary (EMC), which is also the reason which caused the mass extinction event of archaeocyathids and redlichiid/olenellid trilobites (ROECE) [73], such as volcanically [74,75,76] or eustatically [76,77,78] associated redox, carbon negative excursions [73], and the oligotrophic environment caused by aggravated N loss, as well as enhanced P input [78]. Hurdiids thrived from the Wuliuan onwards [79], which may indicate their interspecific competitive relationship with amplectobeluids of the same ecological niche.

According to the pattern of the paleogeographic distribution of amplectobeluids, this group occurred in South China and Laurentia from Cambrian Stage 3 to Drumian, restricted to the subtropical to tropical belt (Figure 7). Thus, in contrast to hurdiids [79], amplectobeluids have the preference for warm water, as seen in anomalocaridids and tamisiocaridids [49,80], which may be controlled by changes in sea temperatures and climate zones.

## 5. Conclusions

New material of “*Anomalocaris*” *kunmingensis* uncovers new characteristics of this species: the most proximal enlarged endite in the distal articulated region and the distal-most secondary-enlarged peduncular endite; elongated endites in the distal articulated region; several distally-swelled anterior auxiliary spines and one posterior auxiliary spine protruding from endites with an acute angle. These characteristics prompt a reassignment of “*Anomalocaris*” *kunmingensis* to a new genus, *Guanshancaris* gen. nov., and support its amplectobeluids affinities. Moreover, the co-occurrence of the frontal appendages of *Guanshancaris kunmingensis* comb. nov. and fragments of trilobites and brachiopods with embayment injury may indicate the durophagous eating habit of *Guanshancaris kunmingensis,* and enhance the possibility of amplectobeluids as a durophagous predator, as well as adding to the understanding of its feeding ecology. In addition to the improved morphological information provided by the new specimens, the distribution date of Amplectobeluidae suggests that both the variety and quantity of this group reached its maximum in the Early Cambrian and apparently decreased in the Middle Cambrian, which may be caused by geochemical and eustatic changes added by volcanic activity. From the spatio-temporal and paleoenvironmental distribution of amplectobeluids, this group prefers shallow water in tropical/subtropical regions.

## Figures and Tables

**Figure 1 biology-12-00583-f001:**
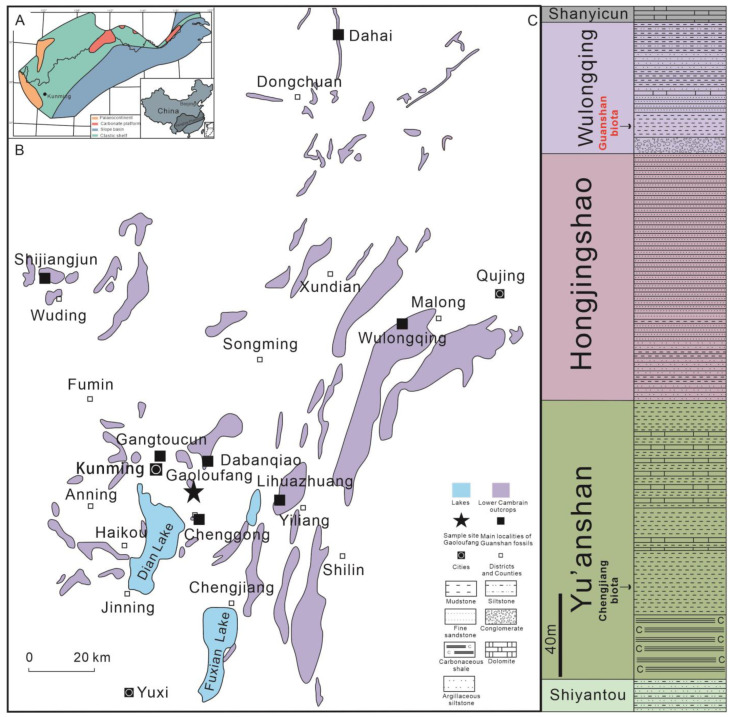
Map of the main fossil localities and stratigraphic column of the Guanshan Lagerstätte in eastern Yunnan Province, South China. (**A**), map of Yangtze Platform in China; (**B**), localities of the Guanshan biota and distribution of lower Cambrian outcrops in eastern Yunnan, South China, including the Gaoloufang section, the sampling site in this paper; (**C**), the stratigraphic column of the Shiyantou, Yu’anshan, Hongjingshao, and Wulongqing formations in the Kunming area ([45], Figure 13), which shows the stratigraphic interval yielding our materials of the Guanshan biota and Chengjiang biota. Fossil localities are from Hu et al. ([28], Figure 3).

**Figure 2 biology-12-00583-f002:**
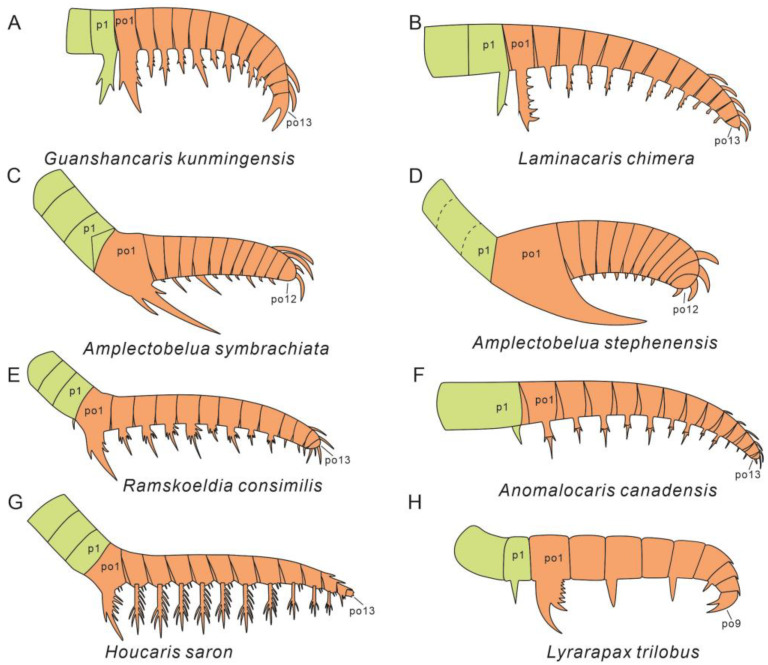
Reconstructions of frontal appendages of *Guanshancaris kunmingensis* (Wang et al., 2010) comb. nov. from the Guanshan biota, and other species with similar morphology. (**A**), *Guanshancaris kunmingensis* (Wang et al., 2010) comb. nov.; (**B**), *Laminacaris chimera*; (**C**), *Amplectobelua symbrachiata*; (**D**), *Amplectobelua stephenensis*; (**E**), *Ramskoeldia consimilis*; (**F**), *Anomalocaris canadensis*; (**G**), *Houcaris saron*; (**H**), *Lyrarapax trilobus*. (**B**,**E**,**F**) redrawn from Pates et al. ([22], Figure 1); (**C**,**D**) redrawn from Pates and Daley ([8], Figure 6); (**G**) redrawn from Wu et al. ([49], Figure 4); (**H**) redrawn from Cong et al. ([56], Figure 3). Colours indicate different parts of the frontal appendage: peduncle (green), the distal articulated region (yellow). P1, the distal-most peduncular podomere; po1 and po13, the proximal-most podomere and terminal podomere in the distal articulated region.

**Figure 3 biology-12-00583-f003:**
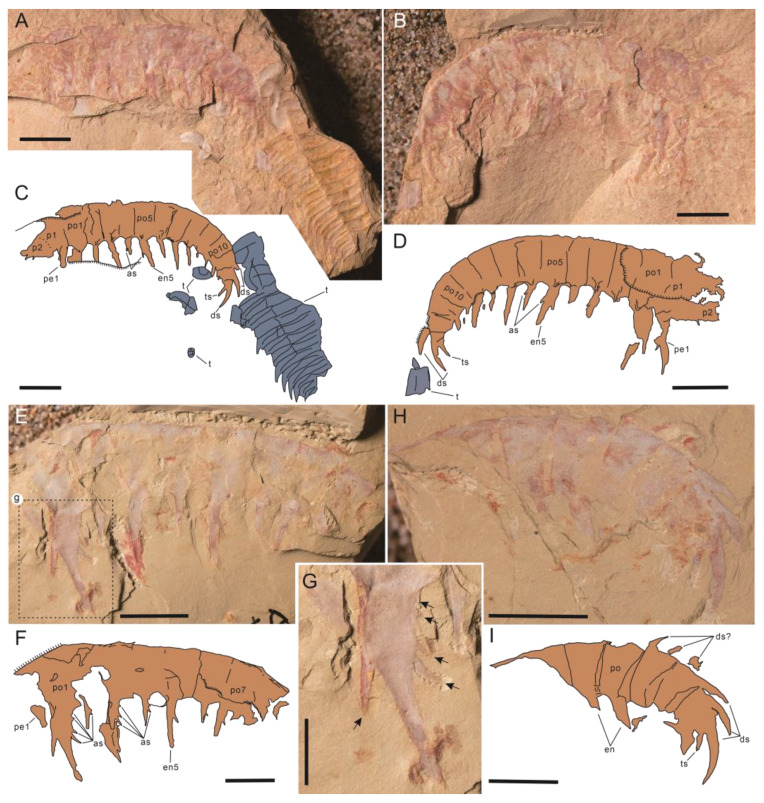
*Guanshancaris kunmingensis* (Wang et al., 2013) comb. nov. from the Guanshan biota, South China (Series 2, Stage 4). (**A**,**B**), part and counterpart, respectively, of holotype, WLQ GLF 02. (**C**,**D**), interpretative drawings of WLQ GLF 02. (**E**), WLQ GLF 05. (**F**), interpretative drawing of WLQ GLF 05. (**G**), close-up of endite on po1 in distal articulated region (boxed in (**E**)), showing anterior and posterior auxiliary spines (black arrows in (**G**)). (**H**), WLQ GLF 10. (**I**), interpretative drawing of WLQ GLF 10. Abbreviations: p—peduncle podomere, po—podomere in distal articulated region, pe—peduncle endite, en—endite in po, as—auxiliary spine, ds—dorsal spine, ts—terminal spine, t—trilobite. All scale bars represent 10 mm, except (**G**), which is 5 mm.

**Figure 4 biology-12-00583-f004:**
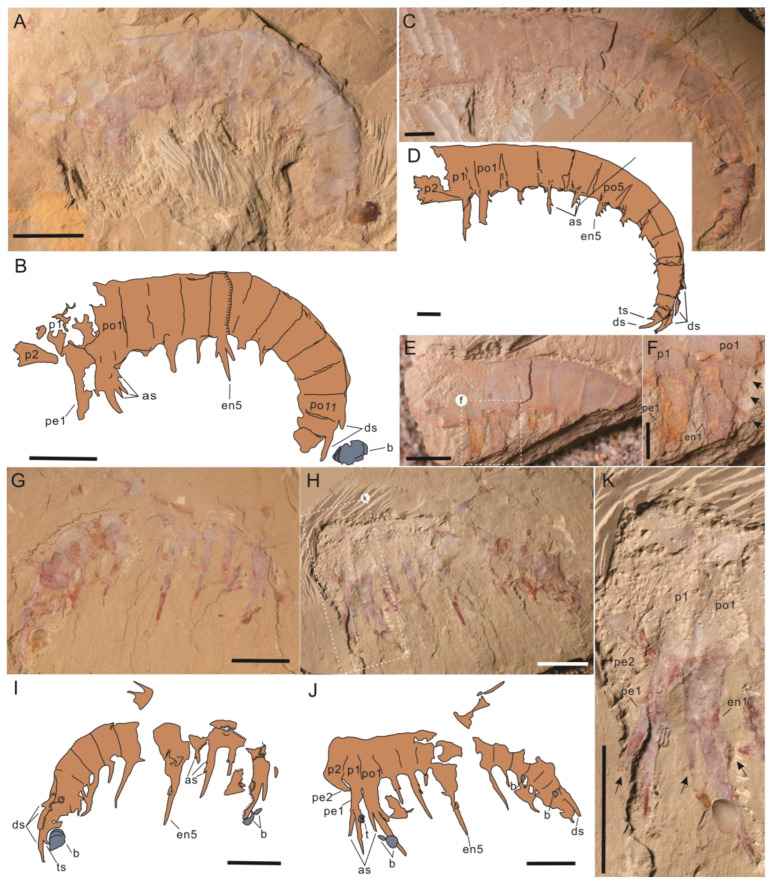
Comparison of frontal appendages in *Guanshancaris kunmingensis* (Wang et al., 2013) comb. nov. and *Laminacaris chimera*. (**A**,**B**,**G**–**K**), *Guanshancaris kunmingensis* (Wang et al., 2013) comb. nov. from the Guanshan biota, South China (Series 2, Stage 4). (**A**), WLQ GLF 08; (**B**), interpretative drawing of WLQ GLF 08; (**G**,**H**), part and counterpart, respectively, of WLQ GLF 03; (**I**,**J**), interpretative drawings of WLQ GLF 03; (**K**), close-up of endites on p1 and po1 (boxed in (**H**)), showing possible endite on p2, posterior auxiliary spine on pe1, anterior and posterior auxiliary spines on en1 (black arrows in (**K**)). (**C**–**F**), *Laminacaris chimera* from the Chengjiang biota, South China (Series 2, Stage 3). (**C**), SK2211B; (**D**), interpretative drawing of SK2211B; (**E**), SJZ B02 454B; (**F**), close-up of endites on p1 and po1 (boxed in (**E**)), showing anterior auxiliary spines on en1 (black arrows in (**F**)). Abbreviations: b—brachiopod, others as in Figure 3. Scale bars represent: 10 mm in (**A**–**D**,**G**–**J**); 5 mm in (**E**,**K**); 2 mm in (**F**).

**Figure 5 biology-12-00583-f005:**
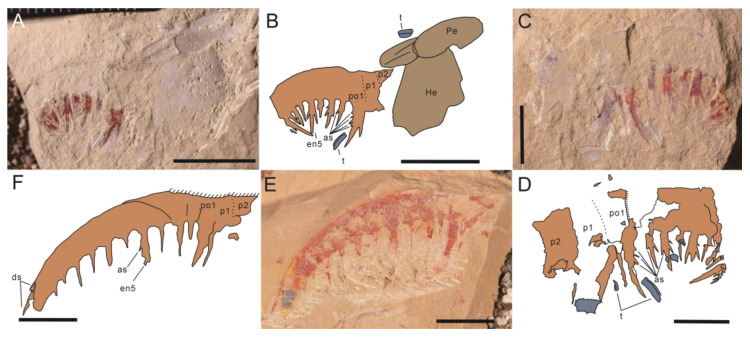
Two small size specimens of *Guanshancaris kunmingensis* (Wang et al., 2013) comb. nov. from the Guanshan biota, South China (Series 2, Stage 4). (**A**,**C**), part and counterpart, respectively, of GLF WLQ 04; (**B**,**D**), interpretative drawing of GLF WLQ 04; (**E**), GLF WLQ 07; (**F**), interpretative drawing of GLF WLQ 07. Abbreviations: Pe—P-element, He—head shield, for others refer to Figure 3 and Figure 4. Scale bars represent: 10 mm in (**A**,**B**); 5 mm in (**C**–**F**).

**Figure 6 biology-12-00583-f006:**
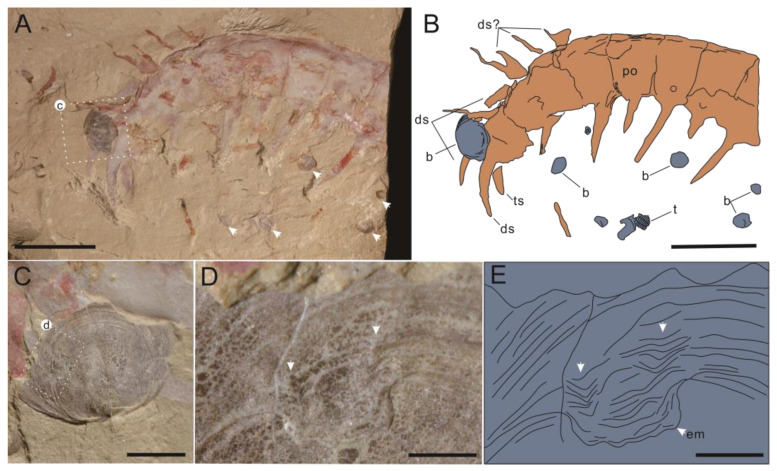
Repaired durophagous shell damages preserved in the brachiopod attached with the frontal appendages of *Guanshancaris kunmingensis* (Wang et al., 2013) comb. nov. (**A**), GLF WLQ 09, showing trilobites and brachiopods associated with frontal appendage (white arrows); (**B**), interpretative drawing of GLF WLQ 09; (**C**), close-up of the injured brachiopods beside the frontal appendage (boxed in (**A**)); (**D**), close-up of fracture embayment in injured brachiopod (boxed in (**C**)); (**E**), interpretative drawing of fracture embayment, showing two drape-like arcs (white arrows in (**D**,**E**)). Abbreviations: em—fracture embayment, for others refer to Figure 3, Figure 4 and Figure 5. Scale bars represent: 10 mm in (**A**,**B**); 5 mm in (**C**); 2 mm in (**D**,**E**).

**Figure 7 biology-12-00583-f007:**
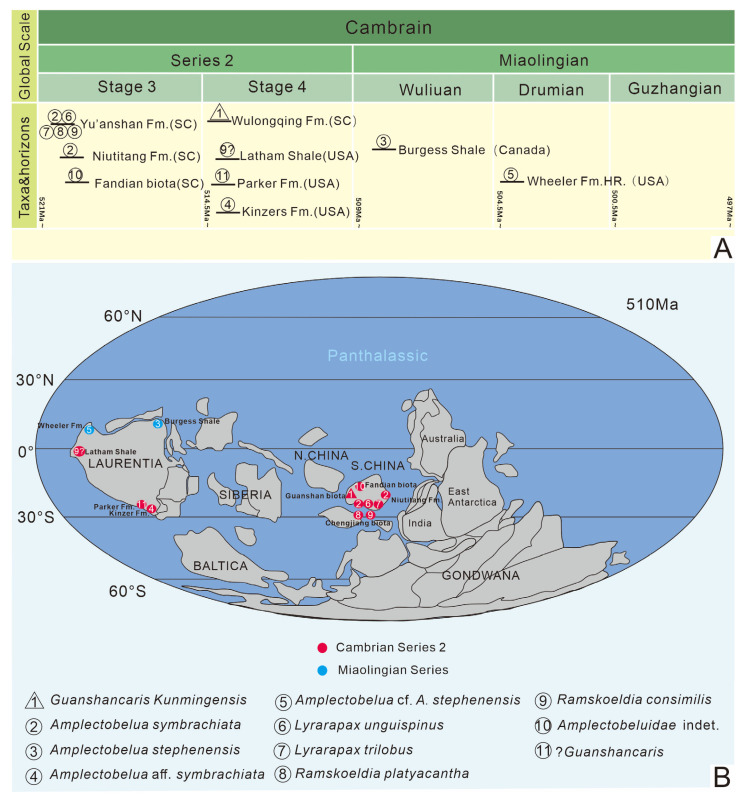
Distribution of the family Amplectobeluidae during the Cambrian. (**A**), Stratigraphical distribution showing amplectobeluids are restricted to Cambrian Series 2 and Miaolingian. (**B**), Paleobiogeographical distribution, indicating that amplectobeluids occur across South China and Laurentia within a relatively narrow tropical belt. Paleocontinental reconstructions during the Early Cambrian time redrawn, modified, and simplified by Torsvik and Cocks ([71], Figure 2.7, 2.8). Each taxon is indicated by specific number. Numbers in (**A**) correspond to those plotted in (**B**). References for different amplectobeluid radiodont taxa: 1 = [7] and this study; 2 = [24,65]; 3 = [12]; 4 = [8]; 5 = [69]; 6 = [53]; 7 = [56]; 8, 9 = [25]; 9? = [22]; 10 = [66]; 11 = [54,69].

**Table 1 biology-12-00583-t001:** Comparison of frontal appendage characteristics of selected radiodont genera.

	Morphology of Frontal Appendage						Morphology of Largest en.					References
	# pd in Peduncle	# pd in d.a.r.	t. m. between pd	# Rows En.	En. in d.a.r. Alternate Long/Short	en. on pd5 Longer than pd3 in d.a.r.	Aux. Paired	Aux. on Both Anterior and Posterior	Aux. Increase Distally Along En.	# Large Aux.	Small Aux. between Large Aux.	
*Guanshancaris kunmingensis*	2	13	Yes	2	Yes	Yes	No	Yes	Yes	2	No	This study [7,10]
*Laminacaris chimera*	2	13	Yes	1?	Yes	Yes?	No	Yes	Yes	5	Yes	[47]
*Amplectobelua symbrachiata*	3	12	Yes	2	Yes	Yes	No	Yes	No	2	No *	[24]
*Amplectobelua stephenensis*	3	12	Yes	2	Yes	Yes	No	No	No	0	No *	[12]
*Ramskoeldia consimilis*	3	13	Yes	2	Yes	Yes?	Yes	Yes	No	2	No	[25]
*Anomalocaris canadensis*	1	13	Yes	2	Yes	No	Yes	Yes	No	2	No *	[15]
*Houcaris saron*	3	13	Yes	2	Yes	No	Yes	Yes	No	2	No	[49]

* Only one aux. on both the anterior and posterior largest endite of *Amplectobelua symbrachiata* and *Anomalocaris canadensis*; auxiliary spines attach to elongated blades that project from the base of the endite and lie directly adjacent to largest endite of *Amplectobelua stephenensis*; pd—podomere; En.—ventral endite; Aux.—auxiliary spine; d.a.r.—distal articulated region; t.m.—triangular membrane. #—the number of each item; ?—indetermination of result.

**Table 2 biology-12-00583-t002:** The occurrences of amplectobeluid radiodonts from worldwide Konservat-Lagerstätten representing different sedimentary environments.

Deposits	Age	Fossil Zones	Strata	Taxa	Body Parts	Abundance	Environment	References
Chengjiang biota, Yunnan, South China	Cam. Stage 3	*Yunnanocephalus* Subzone, *Eoredlichia–Wutingaspis* Zone	Yu’anshan Mb., Chiungchussu Fm.	*Amplectobelua symbrachiata*	HS, OC, GLS, FA, FL	1021	a proximal offshore to lower shoreface setting	[24,25,44,47,53,55,56,64]
				*Lyrarapax unguispinus*	HS, OC, EY, FA, FL, GT, TF, TK, NP, MS	4		
				*L. trilobus*	HS, OC, EY, FA, FL, GT, TK	2		
				*Ramskoeldia platyacantha*	HS, OC, GLS, FA, FL	5		
				*R. consimilis*	HS, OC, GLS, FA, FL	5		
Niutitang Fm., Hunan South China	Cam. Stage 3	*Parabadiella–Mianxiandiscus* Interval Zone	Niutitang Fm.	*Amplectobelua symbrachiata*	FA	1	an offshore shelf-basin facies of the Yangtze Platform; a marginal back-arc basin	[65,68]
Fandian biota, Sichuan, South China	Cam. Stage 3	*Yilangella–Zhangshania* Zone	Yuxiansi and Jiulaodong Fm.	Amplectobelluidae indet. 1	FA	1	an offshore to shoreface continental platform	[66]
Guanshan biota, Yunnan, South China	Cam. Stage 4	*Palaeolenus* Zone, *Megapalaeolenus* Zone	Wulongqing Fm.	*Guanshancaris kunmingnesis*	FA, HS	149	an offshore transition between fair-weather wave base and storm wave base	[7,10,28]; this study
Kinzers Fm., Pennsylvania, USA	Cam. Stage 4	*Bonnia–Olenellus* Zone	Kinzers Fm.	*Amplectobelua* aff. *symbrachiata*	FA	3	seaward of a carbonate shelf	[8]
Parker Quarry Lagerstätte, northwest Vermont, USA	Cam. Stage 4	*Bolbolenellus euryparia* Zone or overlying *Nephrolenellus multinodus* Zone	Parker Fm.	?*Guanshancaris*	FA	1	seaward margin of a carbonate platform with a steep slope; part of the deep-water Franklin Basin	[54,69]
Latham Shale, California, USA	Cam. Stage 4	*Bristolia mohavensis–Peachella iddingsi* Zones	Latham Shale, Marble and Providence Mountains	*Ramskoeldia consimilis*?	FA	4	a proximal shelf above the fair-weather wave	[22]
Burgess Fm. British Columbia, Canada	Cam. Wuliuan	Latest *Glossopleura* Zone to early–middle *Bathyuriscus–Elrathina* Zone	’Thick Stephen’ (Burgess Shale) Fm.	*Amplectobelua stephenensis*	FA	5	seaward margin of a carbonate platform with a steep slope; below storm wave base	[12]
Wheeler Fm. House Range, Utah, USA	Cam. Drumian	*Bolaspidella* trilobite Zone, *Ptychagnostus atavus* agnostoid Zone	Wheeler Fm.	*Amplectobelua* cf. *A. stephenensis*	FA	1	offshore, outer shelf, adjacent to a carbonate ramp; deep-water settings; below storm wave base	[69]

Abbreviations: Cam.—Cambrian; HS—head shield; GLS—gnathobase–like structures; EY—eye; FA—frontal appendage; FL—body flap; GT—gut; OC—oral cone; TF—tail fan;
TK—trunk; NP—neuropil; MS—muscle. ?—indetermination of genus or species.

## Data Availability

Not applicable.

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
