# Peer review of "Amplectobeluid Radiodont Guanshancaris gen. nov. from the Lower Cambrian (Stage 4) Guanshan Lagerstätte of South China: Biostratigraphic and Paleobiogeographic Implications"

_biology, 2023, doi:10.3390/biology12040583_

Round 1
Reviewer 1 Report
In the paper “Amplectobeluid radiodont Guanshancaris gen. nov. from the lower Cambrian (Stage 4) Guanshan Lagerstätte of South China: Biostratigraphic and paleobiogeographic implications” by Zhang et al., authors based on new material redescribe previously known as “Anomalocaris” kumnimgensis species from the Guanshan biota, lower Cambrian (Stage 4). Originally described as Anomalocaris in 2013, it was later on shown that it is closer to the group Amplectobeluidae, and thus should be excluded from the genus Anomalocaris. However, it was not officially removed and renamed, even though this taxon is the most abundant radiodont in the Guashan biota. Authors not only give a new life to this taxon by officially describing it, but also add new valuable characteristics to it. For example, based on the frontal appendage morphology, it was possible to show that this taxon should be placed in its own genera, because it does not fit in any existing genera of the group Amplectobeluidae. Also based on its proximity to broken parts of trilobites and brachiopods, authors proposed the durophagous eating strategy of newly described genera Guanshancaris. After analysis of worldwide distribution of the group Amplectobeluidae, they conclude that this group preferentially was living in low latitude regions and in shallow water environments. This contribution is undoubtedly long-awaited in the realm of early arthropod evolution, especially group Radiodonta. Hence, I am convinced that this paper should be accepted for the publication.
Nevertheless, I do have a number of concerns and comments, which should be addressed/solved before the publication. The line-by-line comments are attached to the review (it is in the form of the highlighted comments inside of the pdf file that was provided for the review). Here, I will just highlight some of the comments from the file.
1. I would strongly recommend adding stratigraphic column to the figure 1. It will be easier to readers to understand where from the material is coming as well as for following researches who would want to use the results which you describe in your paper. There are also not all symbols explained in the legend. In addition, in material and methods section, it is mentioned that Laminacaris is used to comparison. However, from just the description in the text, it is unclear why exactly this taxon was chosen for comparison, as well as how it correspondence to Guanshancaris spatially and temporally. Maybe add this information to the figure too.
2. Even though in the figures 3 and 4 it is possible to see 13 distal podomeres, but in the schematic drawings of figure 2, I was able to count only 12 of them. Also, figure 2 has different colors which are not explained in the caption of figure itself. I would suggest to mark the different parts of frontal appendage morphology in this schematic figure, as well as may be put numbers on the podomeres, so it would be clear from which end counting is happening. Otherwise, it is quite hard to read the text and compare/relate it to the figures. Concerning figure 2, in the text there is also mentioning of the genus Lyrapax, but for some reason in does not shown in the figure 2, although all other mentioned genera are.
3. In the file, which I received for the review, the figure 6 was equal to figure 5, so I am not sure what was shown there.
4. In the remarks to the genus Guanshancaris it is mentioned that “Unlike Lyrarapax and Ramskoeldia, the podomeres of Guanshancaris have a higher height/length ratio (taller rectangular).” – lines 180-181, later on, in the remarks to species it is said: “the height to length ratio of the podomeres averages ∼2.0” – line 317. If that parameter is used for the description of the taxa, why it is not mentioned in the description of the genus, species? Also, in the lines320-321 in the text it is mentioned that appendages from Parker Fm, considered to be Guanshankaris. If so, why they are not included into synonyms of this genus, and why it is mentioned in the remarks to the species, not genera?
5. Text is very nicely written, however, there are missing commas and articles, not correspondence of verbs and nouns, some misspelling of taxa. Some of the papers on the reference list do not have number of volume/issue/pages.
I am sure this paper will become a great contribution to the body of Radiodonta investigations.

Reviewer 2 Report
This MS is well written. The descriptive part is very accurate and well illustrated by photographs and line-drawings. New detailed information is given on the palaeogeographic distribution of several taxa and their ecological preferences. Part of the discussion concerning the feeding mode of the anomalocaridid described in this MS would require more cautiousness (see remarks below). I would recommend the authors to mainly concentrate on the functional morphology of appendages which is one of the best-evidenced part of the MS. I would be happy to recommend the revised version of this MS for publication. Please have the final version of your MS checked by a native speaker of English or an editing company.
Line 69- Lobopod is a morphological term for soft-bodied leg. Please use lobopodians to designate the group
FIG 2- Laminacaris not Laminacairs
Table 1- The first and second lines of the table looks odd. Please check and make sure that the final version will appear correctly.
Line 282 +- remains of various animals frequently occur on rock slabs from Chengjiang and rock slabs. That isolated frontal appendages of anomalocaridids are found associated with trilobites and brachiopods does not prove any feeding relationships. Please add a sentence to explain this problem.
I don’t think the brachiopod shell could have been broken into pieces by these spines.
Line 352- Do you evidence from gut contents ? If so please give more details on the shell fragments found in gut (whole or broken) and the possible feeding mechanism (how prey is caught by appendages and transferred to the oral cone to be eventually ingested- what is the role of the oral cone ?)
Line 366- In Sidneyia, most disarticulated elements belong to juvenile trilobites. It is clear that the gnathobases were used to grasp and break small trilobites into pieces. As for brachiopods, they occur more rarely in Sidneyia’s gut and do not seem to be broken.Perhaps they were swallowed whole ?
Line 372- Very weal support (see above). I would be more careful. The strength of your MS is to analyze the functional morphology of grasping appendages (see your comparative study). The rest is much more speculative and if I were you I would be more cautious.
Line 375- I don’t think so. Do you mean that the animal was preserved in the act of feeding on trilobites and brachiopods. In this case the whole anomalocaridid animal would be there or at least part of the body. Instead the only remains you have are grasping appendages. To me the faunal assemblages shown here result from decay and transportation by bottom currents. I agree with you that it is not a high-energy environment.
Line 378- Repaired damage ? Please delete this sentence. There is no sufficient evidence to prove that this supposed injury was inflicted by this anomalocarid.
Line 385- “the self–repairing behavior of brachiopods may indicate that the injured brachiopod was in the soft–shell stage”. This brachiopod had a shell. There is no stage when the animal is soft (in contrast with molting arthropods). I don’t see any evidence of any shell repair ? Do you mean that the brachiopod was injured during a predator’s attack, survived its injury, repaired its shell and was caught again ? This scenario sounds very odd and is poorly convincing. If you want to discuss the diet of anomalocaridids, consider the gut contents first then also incorporate evidence from the functional morphology of appendages (as you did). Removing over-speculative statements will add more strength to your paper.
Line 397- language problem in this sentence
Line 432- Perhaps the ecological preferences have also something to do with the food availability. Perhaps you can mention it. Predators occur where sufficient food is.
